# Screening for anxiety in patients with cancer: Diagnostic accuracy of GAD-7 items considering lowered GAD-7 cut-offs

Miriam Grapp[1,2]*, Till J. Bugaj[1,2], Valentin Terhoeven[1], Hans-Christoph Friederich[1,3], Imad Maatouk[1,4]

1 Department of General Internal and Psychosomatic Medicine, University Hospital Heidelberg, Heidelberg, Germany, 2 National Center for Tumour Diseases (NCT), University Hospital Heidelberg, Heidelberg, Germany, 3 DZPG (German Centre for Mental Health–Partner Site Heidelberg/ Mannheim/ Ulm), Heidelberg, Mannheim, Ulm, Germany, 4 Section of Psychosomatic Medicine, Psychotherapy and Psychooncology, Department of Internal Medicine II, Julius-Maximilian University Würzburg, Würzburg, Germany

* miriam.grapp@med.uni-heidelberg.de

**Data Availability Statement:** The data set used in this study is available on the Figshare data repository: https://figshare.com/articles/dataset/Screening_for_anxiety_in_patients_with_cancer_

## Abstract

### Background

A standard questionnaire for generalized anxiety disorders is the GAD-7. Attempts to improve its screening capacity in oncological settings resulted in a discussion about lowering its cut-off. This study examines the diagnostic accuracy of the GAD-7 items depending on applied cut-offs and whether, similar to depressive symptoms, a distinction between somatic-emotional and cognitive items might be relevant.

### Patients and methods

Screening data from 4705 patients with cancer who were treated at the outpatient clinic of the National Centre for Tumour Diseases in Heidelberg were analysed. For the individual GAD-7 items sensitivity, specificity, positive and negative predictive values, and Clinical Utility Index were determined for cut-off $\geq 7$, $\geq 8$, $\geq 10$ and $\geq 15$ in the GAD-7 questionnaire.

### Results

The best overall diagnostic accuracy was found for a cut-off $\geq 8$. The cognitive items had the best diagnostic accuracy for identifying severe GAD (cut-off $\geq 15$), and the somatic-emotional items had the best diagnostic accuracy for identifying mild to moderate GAD (cut-off $\geq 7$, $\geq 8$ and $\geq 10$).

### Conclusions

Our data support the recommendation of lowering the GAD-7 cut-off in oncology settings and suggest that in anxiety disorders, a symptom overlap between the physical illness and a possible mental disorder should be considered.

diagnostic_accuracy_of_GAD-7_items_
considering_lowered_GAD-7_cut-offs/27880509/
1?file=50690409 (https://doi.org/10.6084/m9.
figshare.27880509.v1).

**Funding:** For the publication fee we acknowledge
financial support by Heidelberg University.

**Competing interests:** The authors have declared
that no competing interests exist.

## Introduction

High degrees of anxiety in patients with cancer are comprehensible and attributable to the devastating diagnosis and often protracted treatment. In the context of a cancer diagnosis, anxiety can therefore be considered as an adaptive reaction with functional character that, for example, motivates patients to attend examination and treatment appointments or to seek social support. However, anxiety experienced by patients often goes far beyond this functional level and thus strongly influences both quality of life and disease progression [1]. For instance, anxiety is a relevant predictor of anticipatory nausea [2], and is negatively related to treatment adherence [3]. With prevalence rates of 9.8–19% [4–7], anxiety disorders are as frequent as depressive disorders in oncology patients [8, 9]. However, empirical evidence suggests that anxiety as a comorbidity of cancer is underdiagnosed and undertreated in clinical practice [10, 11].

A standard screening instrument for generalized anxiety disorder (GAD) according to the DSM-5 criteria is the GAD-7 [12, 13]. Attempts to improve its screening capacity in oncological settings resulted in the discussion about its optimal cut-off: While the American Society for Clinical Oncology (ASCO) recommends a cut-off $\geq 10$ indicating the need for intervention [14], a meta-analysis of Plummer et al. [15] suggested a significantly lower cut-off ($\geq 8$). In a comprehensive study on a representative sample of 2142 patients with cancer, Esser et al. [16] concluded that the best ratio between sensitivity and specificity is at an even lower cut-off ($\geq 7$). Another important aspect is that in some studies on depressive disorders in patients with cancer, a symptom overlap with somatic disease is reported [17–19], and the question of whether somatic items are appropriate for screening depressive disorders has been the subject of controversy [20–23]. The GAD-7 also includes items that might be assigned to a more physical (i.e. somatic) symptom dimension (e.g. 'trouble relaxing') and items that might be assigned mainly to a cognitive symptom dimension (e.g. 'not being able to stop/control worrying'). However, there is no such clear and explicit distinction for GAD-7 items as for example in depression screening with PHQ-9 [24] and a discussion about the relevance of somatic items in the identification of anxiety disorders in cancer patients has not yet emerged. Therefore, one aim of the present study is to find out whether a distinction between a somatic-emotional and a cognitive symptom dimension could be relevant in the screening of generalized anxiety disorder in patients with cancer. Thus, the objective of our study is not to evaluate the quality criteria of the GAD-7 questionnaire itself; rather, our focus is on the individual symptoms of generalized anxiety disorder as assessed by the GAD-7 items. Given that some anxiety symptoms are also present in individuals with cancer who do not have generalized anxiety disorder, this analysis provides an opportunity to examine whether the conventional symptoms of generalized anxiety disorder may lack diagnostic specificity in an oncological context. Additionally, we explore whether diagnostic accuracy might be enhanced by adjusting the GAD-7 cut-off. Analogous to a methodological approach of Mitchell et al. [24] and Grapp et al. [25] for the analysis of PHQ-9 depression items, this study investigates (1) whether the diagnostic accuracy of GAD-7 items changes depending on the GAD-7 cut-off and (2) whether a distinction between somatic and cognitive items may also be relevant in the screening of GAD in oncological settings.

## Materials and methods

### Participants and procedure

Our sample consisted of patients who had started chemotherapy at the University Hospital Heidelberg. Routine data was collected as part of a standardized bio-psycho-social distress screening between 2011 and 2018, as this period was characterized by a consistent screening

methodology and standardized data collection protocols. After 2018, changes in the screening process, modifications in the variables collected, and a prolonged interruption of screening activities due to the COVID-19 pandemic introduced significant variations. Therefore, the selected timeframe ensures the comparability and reliability of the data.

German-speaking patients with a cancer diagnosis (regardless of stage and severity of tumour disease) and aged at least 18 years were enrolled in the screening. Patients with severe physical or cognitive impairments affecting their ability to participate in the screening and give informed consent were excluded. After patients gave their oral informed consent to participate in the screening procedure, they completed several screening instruments for psychological stress, including the GAD-7, as well as a socio-demographic questionnaire. The GAD-7 data and the necessary socio-demographic data were analysed retrospectively. The screening data for this research purpose was accessed on December 15, 2018. During or after data collection, the authors had no access to information that could identify individual participants. The study was carried out in accordance with the Declaration of Helsinki. Ethical approval was granted by the ethics committee of the University of Heidelberg (S-753/2018).

### Measures and criterion references

Sociodemographic and disease-related data were retrieved from the clinics' patient documentation system. The German version of the GAD-7 was used to screen for anxiety disorders. Although initially developed as a screening tool for Generalized Anxiety Disorder, the GAD-7 also demonstrates acceptable sensitivity and specificity for identifying other anxiety disorders (Panic Disorder, Social Anxiety, and PTSD) [15]. In the present study, however, the focus was specifically on the symptoms of generalized anxiety disorder. Consequently, the diagnostic accuracy of the GAD-7 was examined solely in relation to generalized anxiety disorder, and other anxiety disorders were not considered.

According to recent research and in line with the objective of our study, we used four different cut-offs for the identification of GAD: the standard cut-off $\geq 10$ (ASCO) [14], two lowered cut-offs $\geq 8$ and $\geq 7$ according to the recommendations of Plummer et al. [15] and Esser et al. [16], and a cut-off $\geq 15$, corresponding to the presence of severe GAD. To investigate whether a distinction between a somatic-emotional and a cognitive symptom dimension might be relevant in screening for generalized anxiety disorder in cancer patients, we needed to classify the GAD-7 items into these two dimensions. To the best of our knowledge, such a categorization did not exist before and therefore there was no preliminary work that we could refer to. We therefore used expert consensus for the categorization: seven raters (clinical/scientific experts; psycho-oncologists, physicians) independently assigned each item to one the two symptom dimensions. The classification was determined by the level of agreement between the raters, with items assigned by majority vote when full consensus was not achieved. The resulting categorization used for our analysis is shown in Table 1.

As the data were collected as part of a routine screening process, there is no external, and therefore independent, criterion that can be used to validate the GAD-7 score. It is therefore not possible to completely eliminate the possibility of partial self-correlation with GAD-7 items and the GAD-7 score. Therefore, the risk that the diagnostic accuracy measures used in this study may be overestimated or biased must be considered when analyzing and interpreting the results.

### Statistical analyses

This study used the statistical approach of Grapp et al. [25] and the description of statistical analyses partly reproduces their wording. Statistical analysis was performed with IBM SPSS,

**Table 1. Categorisation of the GAD7 items into somatic-emotional or cognitive.**

| somatic-emotional | |
|---|---|
| Item 1 | 'feeling nervous, anxious, or on edge' |
| Item 4 | 'trouble relaxing' |
| Item 5 | 'being so restless that it is hard to sit still' |
| Item 6 | 'becoming easily annoyed or irritable' |
| Item 7 | 'feeling afraid as if something awful might happen' |
| cognitive | |
| Item 2 | 'not being able to stop or control worrying' |
| Item 3 | 'worrying too much about different things' |

version 27. Descriptive statistics were calculated of the sociodemographic and disease-related characteristics. Receiver operating characteristics (ROC) were used to define the optimal cut-off for the individual items. The following measures were calculated for the items' diagnostic accuracy: sensitivity (Se), specificity (Sp), positive and negative predictive value (PPV and NPV), and Clinical Utility Index (CUI). The CUI is a composite score of diagnostic accuracy and subdivided into the positive utility index (UI+ = sensitivity x PPV), which measures case-finding accuracy, and the negative utility index (UI- = specificity x NPV), which measures screening accuracy [24, 26].

## Results

### Sample and item characteristics

The total sample comprised 4705 patients, whose baseline data are presented in Table 2 together with the prevalence of GAD according to the different GAD-7 cut-offs. The prevalence of GAD according to the ASCO recommendations (cut-off $\geq$ 10) was 15.3%, and the prevalence of GAD with a lower GAD-7 cut-off was 22.3% (cut-off $\geq$ 8) and 28.2% (cut-off $\geq$ 7).

The frequencies and mean scores of the GAD symptoms (represented by the individual GAD-7 items) depending on different GAD-7 cut-offs as well as for the total sample are provided in Table 3. For all GAD-7 cut-offs and the total sample, the somatic-emotional items 1 (nervousness) and 4 (restlessness) showed the highest incidence and mean scores. While at lower cut-offs the incidence and mean scores of the cognitive items 2 (inability to stop worrying) and 3 (excessive worry) tended to be in a middle range, at a cut-off $\geq$ 15 these two items were at an equivalent level to the two high-scoring somatic items 1 and 4.

### Diagnostic accuracy of the GAD-7 items

The ROC curves of the individual GAD-7 items revealed the best discrimination ability for all items at a cut-off $\geq$ 2. Therefore, this cut-off was chosen for further analyses. In Table 4 all measures of diagnostic accuracy for the individual GAD-7 items are provided.

At a cut-off $\geq$ 10, sensitivity for all items was in a poor to acceptable (25.0–77.8%), specificity in a good (86.5–95.4%), PPV in a poor (34.1–45.1%), and NPV in a good range (91.5–97.0%). At a lowered cut-off ($\geq$ 7 or $\geq$ 8), there was a marginal decrease in sensitivity (21.3–72.8%) and an increase in specificity (95.0–99.5%). The PPV increased to an acceptable to good range (76.9–97.2%), while the NPV slightly decreased but remained in an acceptable to good range (76.1–92.4%). At a cut-off $\geq$ 15, sensitivity was in a good range (91.7–99.1%) for all items except for the two items 4 and 5. For all items, specificity (83.9–95.3%) and NPV (97.1–99.9%) was good, while PPV was poor (24.2–39.8%). Of all items, the two somatic-

**Table 2. Demographic data, clinical data and prevalence of GAD depending on different GAD-7 cut-offs.**

| Gender | | *n* | *%* |
|---|---|---|---|
| | Male | 2005 | 42.6 |
| | Female | 2700 | 57.4 |
| **Age** in years | | *Mean* | *SD* |
| | | 57.62 | 13.17 |
| **Primary cancer site** | | *n* | *%* |
| | Breast | 1160 | 24.7 |
| | Colorectal | 483 | 10.3 |
| | Skin | 445 | 9.5 |
| | Pancreas | 416 | 8.8 |
| | Gastrointestinal | 315 | 6.7 |
| | Female genitalia | 309 | 6.6 |
| | other digestive organs | 274 | 5.8 |
| | Male genitalia | 245 | 5.2 |
| | Urinary tract | 161 | 3.4 |
| | Haematological | 110 | 2.3 |
| | Respiratory tract | 84 | 1.8 |
| | Bone and soft tissue | 79 | 1.7 |
| | Others | 400 | 8.5 |
| | Missing values | 224 | 4.8 |
| **Time since diagnosis** in month | | *Mean* | *SD* |
| | | 5.52 | 6.21 |
| **Prevalence of generalized anxiety disorder (GAD)** according to GAD-7 | | *n* | *%* |
| | GAD-7 cut-off $\geq$ 7 | 1327 | 28.2 |
| | GAD-7 cut-off $\geq$ 8 | 1046 | 22.3 |
| | GAD-7 cut-off $\geq$ 10 (moderate GAD) | 724 | 15.4 |
| | GAD-7 cut-off $\geq$ 15 (severe GAD) | 231 | 4.9 |

**Table 3. Prevalence (%) and mean scores (mean (SD)) of the GAD-7 items for different GAD-7 cut-offs.**

| Item | cut-off $\geq$ 7 % Mean (SD) | cut-off $\geq$ 8 % Mean (SD) | cut-off $\geq$ 10 % Mean (SD) | cut-off $\geq$ 15 % Mean (SD) | total sample % Mean (SD) |
|---|---|---|---|---|---|
| 'feeling nervous, anxious, or on edge' | 62.2 1.93 (0.86) | 72.8 2.12 (0.84) | 84.4 2.36 (0.71) | 99.1 2.87 (0.37) | 20.1 1.01 (0.88) |
| 'not being able to stop or control worrying' | 40.1 1.49 (0.88) | 49.4 1.65 (0.91) | 63.1 1.90 (0.91) | 94.4 2.59 (0.63) | 12.6 0.69 (0.83) |
| 'worrying too much about different things' | 47.1 1.63 (0.87) | 57.1 1.80 (0.87) | 70.0 2.03 (0.85) | 92.6 2.62 (0.61) | 14.8 0.76 (0.86) |
| 'trouble relaxing' | 62.2 1.90 (0.86) | 72.8 2.09 (0.83) | 84.7 2.34 (0.76) | 99.1 2.84 (0.39) | 19.9 0.89 (0.93) |
| 'being so restless that it is hard to sit still' | 21.3 0.94 (0.87) | 25.7 1.02 (0.93) | 31.9 1.15 (0.96) | 46.85 1.48 (1.06) | 6.7 0.38 (0.67) |
| 'becoming easily annoyed or irritable' | 35.1 1.32 (0.89) | 40.7 1.42 (0.93) | 47.9 1.54 (0.97) | 68.4 1.98 (0.99) | 11.8 0.60 (0.78) |
| 'feeling afraid as if something awful might happen' | 45.4 1.56 (0.92) | 55.1 1.72 (0.94) | 68.9 1.99 (0.91) | 91.3 2.51 (0.72) | 13.6 0.68 (0.85) |

**Table 4. Diagnostic accuracy of GAD-7 items depending on different GAD-7 cut-offs.**

| cut-off | Se (95%-CI) | Sp(95%-CI) | PPV (95%-CI) | NPV (95%-CI) | UI(+)[a] | UI(-)[a] |
|---|---|---|---|---|---|---|
| Item 1: 'feeling nervous, anxious, or on edge' | | | | | | |
| ≥ 7 | 62.3 (59.6–64.9) | 96.5 (95.8–97.1) | 87.5 (85.3–89.6) | 86.6 (85.4–87.6) | 0.55 | 0.84 |
| ≥ 8 | 72.8 (70.0–75.5) | 95.0 (94.2–95.7) | 80.8 (78.1–83.2) | 92.3 (91.4–93.2) | 0.59 | 0.88 |
| ≥ 10 | 77.5 (73.5–81.1) | 86.5 (85.5–87.6) | 40.5 (37.3–43.7) | 97.0 (96.4–97.5) | 0.32 | 0.84 |
| ≥ 15 | 99.1 (96.9–99.9) | 83.9 (82.8–85.0) | 24.2 (21.5–27.1) | 99.9 (99.8–99.9) | 0.24 | 0.84 |
| Item 2: 'not being able to stop or control worrying' | | | | | | |
| ≥ 7 | 40.2 (37.5–42.8) | 99.1 (98.7–99.2) | 97.2 (95.5–98.4) | 80.7 (79.5–81.9) | 0.39 | 0.80 |
| ≥ 8 | 49.5 (46.4–52.3) | 98.7 (98.3–99.1) | 94.5 (92.2–96.2) | 87.2 (86.1–88.2) | 0.47 | 0.86 |
| ≥ 10 | 48.6 (44.0–53.0) | 92.7 (91.8–93.5) | 43.6 (39.4–47.9) | 94.0 (93.2–94.7) | 0.21 | 0.87 |
| ≥ 15 | 94.4 (90.5–96.9) | 92.5 (91.7–93.3) | 39.8 (35.7–44.0) | 99.6 (99.4–99.8) | 0.38 | 0.92 |
| Item 3: 'worrying too much about different things' | | | | | | |
| ≥ 7 | 47.3 (44.5–50.0) | 97.9 (97.3–98.3) | 90.0 (87.5–92.1) | 82.4 (81.2–83.6) | 0.43 | 0.81 |
| ≥ 8 | 57.4 (54.3–60.4) | 97.3 (96.7–97.8) | 86.0 (83.2–88.2) | 88.8 (87.8–89.8) | 0.49 | 0.86 |
| ≥ 10 | 59.9 (55.4–64.2) | 90.3 (89.4–91.2) | 42.2 (38.5–45.9) | 95.0 (94.3, -95.7) | 0.25 | 0.86 |
| ≥ 15 | 93.0 (88.9–95.9) | 89.1 (88.2–90.0) | 30.8 (27.4–34.4) | 99.6 (99.3–99.7) | 0.29 | 0.89 |
| Item 4: 'trouble relaxing' | | | | | | |
| ≥ 7 | 62.2 (59.5–64.8) | 96.8 (96.2–97.4) | 88.7 (86.5–90.6) | 86.6 (85.4–87.7) | 0.55 | 0.84 |
| ≥ 8 | 72.8 (70.0–75.5) | 95.3 (94.6–96.0) | 81.9 (79.3–84.3) | 92.4 (91.5–93.2) | 0.59 | 0.88 |
| ≥ 10 | 77.8 (73.6–81.4) | 86.9 (85.8–87.9) | 41.2 (38.1–44.5) | 97.0 (96.5–97.6) | 0.32 | 0.84 |
| ≥ 15 | 99.1 (96.9–99.8) | 84.2 (83.1–85.2) | 24.6 (21.8–27.5) | 99.9 (99.8–99.9) | 0.24 | 0.84 |
| Item 5: 'being so restless that it is hard to sit still' | | | | | | |
| ≥ 7 | 21.3 (19.1–23.6) | 99.5 (99.2–99.7) | 90.4 (86.6–93.4) | 76.1 (74.7–77.3) | 0.19 | 0.74 |
| ≥ 8 | 25.7 (23.1–28.5) | 99.1 (98.8–99.4) | 85.9 (81.5–89.6) | 82.2 (81.0–83.3) | 0.22 | 0.81 |
| ≥ 10 | 25.0 (21.2–29.0) | 95.4 (94.7–96.0) | 39.3 (33.8–44.9) | 91.5 (90.6–92.3) | 0.09 | 0.87 |
| ≥ 15 | 46.7 (40.1–53.4) | 95.3 (94.7–95.9) | 34.5 (29.2–40.0) | 97.1 (96.6–97.6) | 0.16 | 0.97 |
| Item 6: 'becoming easily annoyed or irritable' | | | | | | |
| ≥ 7 | 35.2 (32.6–37.8) | 97.3 (96.7–97.8) | 84.1 (80.8–87.0) | 79.1 (77.9–80.4) | 0.29 | 0.77 |
| ≥ 8 | 40.8 (37.9–43.8) | 96.4 (95.8–97.0) | 76.9 (73.1–80.3) | 85.0 (83.8–86.1) | 0.31 | 0.82 |
| ≥ 10 | 38.4 (34.1–42.8) | 91.2 (90.3–92.1) | 34.1 (30.1–38.2) | 92.6 (91.5–93.4) | 0.13 | 0.85 |
| ≥ 15 | 68.7 (62.2–74.6) | 88.5 (87.3–89.5) | 28.5 (24.7–32.4) | 97.6 (97.1–98.1) | 0.19 | 0.89 |
| Item 7: 'feeling afraid as if something awful might happen' | | | | | | |
| ≥ 7 | 45.5 (42.8–48.3) | 98.9 (98.5–99.2) | 94.5 (92.4–96.1) | 82.1 (80.9–83.3) | 0.43 | 0.81 |
| ≥ 8 | 55.2 (52.1–58.2) | 98.2 (97.8–98.6) | 90.2 (87.7–92.4) | 88.4 (87.4–89.4) | 0.49 | 0.87 |
| ≥ 10 | 58.6 (54.1–63.0) | 91.6 (90.7–92.4) | 45.1 (41.2–49.1) | 94.9 (94.2–95.6) | 0.26 | 0.87 |
| ≥ 15 | 91.7 (87.4–94.9) | 90.3 (89.4–91.2) | 33.0 (29.4–36.8) | 99.5 (99.2–99.7) | 0.30 | 0.90 |

[a] CUI-scores are classified as poor (≤ 0.49), satisfactory (≥ 0.49), good (≥ 0.64), and excellent (≥ 0.81) [29].

emotional items 1 and 4 showed the highest sensitivity and NPV. The cognitive item 2 and the somatic-emotional item 5 had the highest specificity. The best PPV was found for the cognitive item 2 and the somatic-emotional item 7. Most items showed low case-finding accuracy (UI+), and only the items 1 and 4 had satisfactory UI+ scores for a cut-off ≥ 7 (UI+ = 0.55) and a cut-off ≥ 8 (UI+ = 0.59). For the GAD-7 cut-offs ≥ 8, ≥ 10, and ≥ 15, all items demonstrated excellent levels of screening accuracy (UI- = 0.81–0.97), while for a cut-off ≥ 7, all items demonstrated good to excellent levels of screening accuracy (UI-: 0.74–0.84).

The UI+ and UI- scores for the single items depending on the respective cut-offs are shown in Fig 1A. The two somatic-emotional items 1 and 4 had the best UI+ scores for the lowered

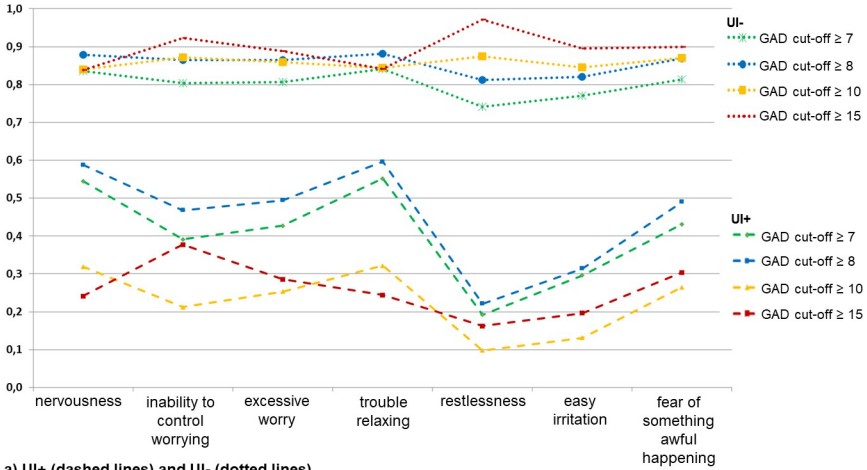

a) UI+ (dashed lines) and UI- (dotted lines)

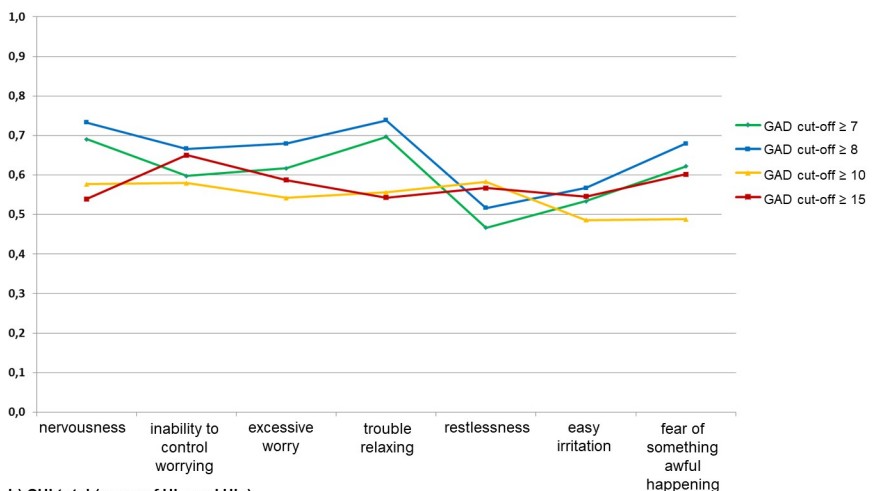

b) CUI total (mean of UI+ and UI−)

**Fig 1. UI+, UI-, and CUI total of single GAD-7 items.** Illustration of the UI+ and UI- scores separately (Fig 1a) and as a global value (CUI total) (Fig 1b), respectively depending on different GAD-7 cut-offs.

GAD cut-offs and for moderate GAD (cut-off $\geq 7$, $\geq 8$, and $\geq 10$) and the best UI- scores for lowered GAD cut-offs (cut-off $\geq 7$ and $\geq 8$). The best UI- scores for moderate and severe GAD (cut-off $\geq 10$ and $\geq 15$) were achieved by the cognitive item 2 and the somatic-emotional item 5. The best UI+ score for severe GAD (cut of $\geq 15$) was also achieved by item 2. Considering UI+ and UI- simultaneously (CUI total), for all items (except item 5), the best ratio between screening and case-finding accuracy results for a cut-off $\geq 8$ (see Fig 1B).

A hierarchy of symptoms from highest to lowest accuracy indicated that for the cut-offs $\geq 7$ and $\geq 8$, the top three were item 4 (difficulty in relaxing), item 1 (nervousness), and item 7 (fear of something awful happening). For a cut-off $\geq 10$, the top three were item 5 (restlessness), item 2 (inability to stop worrying), and item 1 (nervousness), and for a cut-off $\geq 15$, the top three were item 2 (inability to stop worrying), item 7 (fear of something awful happening), and item 3 (excessive worry). Thus, for a cut-off $\geq 15$, primarily the cognitive items displayed the best CUI, while for the lower cut-offs, the somatic-emotional items were the most accurate.

## Discussion

### Main findings

In the present study, the prevalence rate of moderate or severe GAD according to the ASCO recommendations (GAD-7 cut-off $\geq$ 10) was 15.3%, which corresponds to previous studies [4–7]. Our results indicate that, among oncology patients, the optimal balance between sensitivity, specificity, PPV, and NPV of the GAD-7 is achieved with a cut-off score of $\geq$ 8. This aligns with findings from Plummer et al.'s meta-analysis [15] and Esser et al.'s study [16]. Lowering the GAD-7 cut-off in our study significantly increased the percentage of patients who screened positive for generalized anxiety disorder, with rates reaching 28.2% at a cut-off of $\geq$ 7 and 22.3% at $\geq$ 8. We consider these elevated rates to be clinically plausible, given the unique characteristics of our target group, consisting of cancer patients recently diagnosed and at the beginning of their treatment. According to existing literature, anxiety in patients with cancer follows distinct trajectories from diagnosis through the end of first-line treatment to long-term follow-up. Anxiety is particularly prominent in the early stages of diagnosis and at the onset of acute treatment, often reflecting a reaction to the diagnosis and anticipated treatment [4, 27].

The majority of items showed an overall good diagnostic accuracy for all GAD-7 cut-offs. At a cut-off $\geq$ 10, sensitivity for all items was in a poor to medium range, specificity ranged from good to excellent, PPV was in a poor range, and NPV ranged from good to excellent. By lowering the cut-off to $\geq$ 7 or $\geq$ 8, we found a marginal decrease in sensitivity and an increase in specificity. The PPV increased to an acceptable to excellent range, while the NPV slightly decreased but remained in an acceptable to good range. Overall, the best ratio between case finding (UI+ = sensitivity x PPV) and screening accuracy (UI- = specificity x NPV) was found for cut-off $\geq$ 8. We differentiated between items that focused more on a somatic-emotional symptom dimension and items that focused more on a cognitive symptom dimension. The two somatic-emotional items 'nervousness' and 'difficulty in relaxing' achieved the best sensitivity and NPV, and at cut-offs of $\geq$ 7, 8, and 10, the best UI+ and UI scores. Therefore, for detecting and screening cases of mild to moderate GAD, these two items appear to be the most appropriate. For the cognitive item 'inability to stop worrying', the best UI+ score and an excellent UI score were found at the cut-off of $\geq$ 15. Thus, this item seems to play a particularly important role in the identification of severe GAD.

### Clinical implications

Our data showed the best overall diagnostic accuracy for a cut-off $\geq$ 8. This result is in line with the conclusions of Plummer et al. [15] and Esser et al. [16], and emphasizes the recommendation that when using GAD-7 as a screening instrument in oncological settings, its cut-off should be lowered. Lowering the GAD-7 cut-off would lead to earlier identification of anxiety symptoms in cancer patients, which is crucial given their vulnerability to psychological distress. The main aim of a lowered GAD-7 cut-off is to miss as few patients as possible with a potential anxiety disorder and to give as many as possible access to psycho-oncological interventions. This would imply an increase in the number of patients flagged for potential anxiety disorders. These patients should be referred to the psycho-oncological service at an early stage in order to verify the diagnosis of an anxiety disorder in a two-stage process (preferably with a clinical interview).

Depending on the respective cut-off, our results indicated different trajectories of the somatic-emotional and the cognitive GAD-7 items. For the lowered cut-offs, the somatic aspect of inner tension and nervousness seems to be predominant, whereas for the identification of severe GAD (cut off $\geq$ 15), the cognitive aspect of worrying and brooding seems to play

a more important role. Therefore, similar to depressive disorders, one might expect a symptom overlap between the physical illness and a possible mental disorder in anxiety disorders: GAD-related symptoms, e.g., nervousness, agitation or restlessness, can be a consequence of the disease itself or the often lengthy and demanding treatment. Anxiety leads to symptoms of autonomic overactivity, accompanying behavioural responses, altered cognitive processes, and physical symptoms [28]. At the same time, there are endocrine and metabolic changes associated with cancer or its treatment that also affect both physical and psychological states [29]. On the other hand, it should be noted that if somatic symptoms are only attributed to physical illness, this not only leads to the neglect of important psychological aspects in dealing with cancer but also to inadequate efforts in treating the cancer even more aggressively. Therefore, it is important to consider possible symptom overlap in both directions and, especially when patients suffer from severe physical symptoms, to screen systematically for anxiety disorders. Clinicians should ask patients specifically about symptoms of anxiety disorders to initiate a targeted diagnosis and provide patients with appropriate care in accordance with current guidelines.

## Study limitations

The results of the present study are only based on a single questionnaire, and the data were not validated by another questionnaire for anxiety disorders or by a clinical interview. Structured or semi-structured interviews are considered the gold standard for diagnosing anxiety disorders. However, this was not possible in our study, which focused on a large data set. Data were sampled in the context of routine care, where interviews would be too time consuming. It is important to understand that the GAD-7 is simply a screening tool to identify generalized anxiety disorder, but cannot replace a clinical diagnosis. Nevertheless, the GAD-7 is a validated questionnaire showing good consistency with clinical interviews, i.e. in primary care patients [30], in elderly people [31], in psychiatric samples [32], and in patients with various medical conditions, such as coronary artery disease [33], epilepsy [34], or cancer [35]. Given its diagnostic reliability as well as its validity, the GAD-7 is one of the common anxiety measures, both in clinical practice and in research [36]. However, the absence of an additional criterion for evaluating the GAD-7, e.g., HADS as a second screening tool or an interview, results in the criterion standard for this study being the GAD-7 score itself. Thus, the predictor and criterion are not completely independent of each other, which potentially bears the risk that sensitivity and specificity will be overestimated or biased. Therefore, sensitivity and specificity analyses should be interpreted with caution. Further research should therefore incorporate clinical interviews with a sufficiently representative subsample of cancer patients ensure consistency with the GAD-7 results. Some results and conclusions of this paper are based on the categorization of the GAD-7 items into somatic-emotional and cognitive items. As already mentioned in the methods section, such a classification has not yet been made, which means that there is no preliminary work or publications that we could have referred to. Instead, the classification was based on an expert consensus and is to be regarded as a model. The aim of this model was to investigate whether a division into somatic-emotional and cognitive items–in analogy to the extensive discussion regarding the PHQ-9 items–could also be relevant for patients with cancer.

As our data are based on the analysis of a routine questionnaire used in clinical care that focuses on the assessment of psychological distress, no TNM classification was available in the present dataset. Therefore, it was not possible to directly correlate GAD-7 scores with physical disease severity or to perform subgroup analyses. Information on TNM status would allow a more detailed analysis of the possible correlation between anxiety and the cancer stage,

providing a more comprehensive understanding of the relationship between physical and psychological symptom dimensions. Moreover, we did not analyze and interpret our results on a sex or gender-specific basis although there is considerable evidence that there are gender differences in the presentation of anxiety symptoms in cancer patients. In particular, recent studies show that female patients are more anxious than male patients across all tumor entities [37]. Given the complexity of our research question, we chose to limit our scope. We did not address gender differences in anxiety disorders and in the presentation of anxiety symptoms. This could slightly restrict the generalizability of our findings.

It should also be noted that our data was collected between 2011 and 2018 and therefore originates from the period before the COVID-19 pandemic. The COVID-19 pandemic may have introduced new anxiety triggers and altered baseline anxiety levels in recent cancer patients due to increased health-related uncertainties and potential disruptions in social support systems [38]. The data in this study do not account for these pandemic-related influences on the prevalence and significance of anxiety symptoms, which may lead to differences in anxiety profiles between current patients and those in the 2011–2018 dataset, potentially affecting the comparability of findings. Additionally, shifts in mental health awareness and decreasing stigma surrounding anxiety may have influenced self-reporting practices over time. Patients today may feel more comfortable disclosing symptoms than those in earlier years, possibly leading to a slight underreporting of anxiety prevalence in the 2011–2018 dataset.

The strengths of the present study include the fact that our analysis is based on a large sample size, which increases the probability that the sample is broad and yields replicable results. In addition, the use of routine data from everyday clinical practice avoided a major selection bias (with the exception of German language skills) and a high external validity of the results can be postulated. Furthermore, because we did not specify any restrictions on health status or cancer type (or prognosis), our data set can be considered representative.

## Conclusion

Based on a large dataset of routine data, our work makes an important contribution to the investigation of the diagnostic accuracy of GAD-7 items in patients with cancer. In our sample, the individual items showed the best overall diagnostic performance at a GAD-7 cut-off $\geq 8$, which underlines the recommendation of lowering the GAD-7 cut-off in the oncological setting. Furthermore, we observed different trends between the somatic-emotional and the cognitive GAD-7 items. While the cognitive items appear to be more relevant for identifying severe GAD with higher cut-off values, the somatic-emotional items seem to be more relevant for identifying low to moderate GAD. However, it is important to note that these trends have not been statistically tested, so any conclusions should be considered preliminary. Therefore, our data suggest that in anxiety disorders–analogous to depressive disorders–a symptom overlap between the physical illness and a possible mental disorder should be considered. It would be beneficial for future research to consider the stage of the disease, as indicated by the TNM classification. Incorporating the TNM classification into analyses could enhance future research by clarifying the relationship between anxiety symptoms and cancer stage, and by deepening insights into the interplay of physical and psychological symptoms. Another important aspect for further research is a deeper analysis of how GAD-7 scores and the extent of different anxiety symptoms differ between male and female patients with cancer. Given the known gender differences regarding the suffering of anxiety symptoms in oncological patients, this could add an important dimension to our findings.

## Acknowledgments

First and foremost, the authors wish to thank the patients for their willingness to participate in the distress screening. Furthermore, we want to thank the study nurses for their outstanding contribution to the recruitment of patients, and the oncology teams of the National Center for Tumour Diseases and the University Hospital Heidelberg for their support in this study. Finally, we would also like to thank Daniela Wickert for her critical and reflective feedback and proofreading of the manuscript.

## Author Contributions

**Conceptualization:** Miriam Grapp, Valentin Terhoeven, Imad Maatouk.

**Data curation:** Miriam Grapp, Imad Maatouk.

**Formal analysis:** Miriam Grapp, Valentin Terhoeven.

**Investigation:** Miriam Grapp.

**Methodology:** Miriam Grapp, Valentin Terhoeven.

**Project administration:** Miriam Grapp, Imad Maatouk.

**Supervision:** Till J. Bugaj, Valentin Terhoeven, Hans-Christoph Friederich, Imad Maatouk.

**Validation:** Till J. Bugaj, Hans-Christoph Friederich, Imad Maatouk.

**Visualization:** Miriam Grapp.

**Writing – original draft:** Miriam Grapp.

**Writing – review & editing:** Till J. Bugaj, Valentin Terhoeven, Hans-Christoph Friederich, Imad Maatouk.

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
