## [Decision Letter · Decision Letter 0]

11 Oct 2024

PONE-D-24-37266Screening for anxiety in patients with cancer: diagnostic accuracy of GAD-7 items considering lowered GAD-7 cut-offsPLOS ONE

Dear Dr. Grapp,

Thank you for submitting your manuscript to PLOS ONE. After careful consideration, we feel that it has merit but does not fully meet PLOS ONE’s publication criteria as it currently stands. Therefore, we invite you to submit a revised version of the manuscript that addresses the points raised during the review process.

We look forward to receiving your revised manuscript.

Kind regards,

Stephan Doering, M.D.

Academic Editor

PLOS ONE

Reviewers' comments:

Reviewer's Responses to Questions

**Comments to the Author**

1. Is the manuscript technically sound, and do the data support the conclusions?

Reviewer #1: Yes

Reviewer #2: Partly

2. Has the statistical analysis been performed appropriately and rigorously? 

Reviewer #1: Yes

Reviewer #2: Yes

3. Have the authors made all data underlying the findings in their manuscript fully available?

Reviewer #1: Yes

Reviewer #2: Yes

4. Is the manuscript presented in an intelligible fashion and written in standard English?

Reviewer #1: Yes

Reviewer #2: Yes

5. Review Comments to the Author

Reviewer #1: Expand on Methodological Limitations: While the study mentions the lack of validation with another diagnostic measure for anxiety disorders, this limitation could be explored further. Clarifying how the absence of a clinical interview or additional questionnaires might affect the accuracy of the GAD-7 would strengthen the discussion of limitations and transparency.

Consider Including TNM Classification: The study could greatly benefit from incorporating the TNM classification of cancer severity. This would allow for a more detailed analysis of how anxiety levels might correlate with the stage of cancer, offering deeper insights into the physical and psychological interaction.

Elaborate on the Clinical Implications: The conclusion suggests that a lowered GAD-7 cut-off is beneficial for oncological settings. It would be helpful to include more practical guidelines for clinicians on how this change might influence decision-making in clinical practice, particularly regarding referrals to psychosocial support services.

Clarify the Role of Somatic-Emotional and Cognitive Items: The differentiation between somatic-emotional and cognitive items is one of the unique strengths of the study. However, the rationale behind this division could be more fully explained, particularly how this distinction is relevant to the cancer population compared to other populations with anxiety disorders.

Strengthen Data Availability Statement: Although the study provides a link to a Figshare repository for data sharing, it's important to ensure that the repository is fully accessible and includes detailed instructions for others who may want to replicate or further explore the analysis. A clear description of what data is available and how it can be accessed is crucial for PLOS ONE's emphasis on open science.

Consider Exploring Gender Differences Further: The study briefly mentions that the sample is predominantly female. Given that anxiety presentations might differ by gender, a deeper analysis of how GAD-7 scores differ between male and female cancer patients could add an important dimension to the findings.

Reviewer #2: The manuscript by Grapp et al. is an original work that examines the diagnostic accuracy of the items of the GAD-7 scale and, based on this, draws conclusions about the ideal cut-off value of the GAD-7 in cancer patients. It is therefore a clinically relevant and interesting work. The manuscript, which is well written in terms of style, comes from a renowned group in the field of psycho-oncology. Although the manuscript is important, it raises some questions that the authors might consider in the context of a revision:

MAJOR:

Although GAD 7 is named after generalised anxiety disorder, it is also used to screen for other anxiety disorders. In the manuscript, the authors only refer to the accuracy with regard to generalised anxiety disorder. At the very least, the method section should include a justification for not also examining the other anxiety disorders. If necessary, this should be mentioned as a limitation in the discussion.

While a clear distinction is made in the diagnosis of depressive disorders between somatic and non-somatic items (e.g. somatic items of the PHQ-9: ‘feeling tired or having little energy’ or ‘poor appetite or overeating’), this distinction appears not really comprehensible for the GAD-7. It is not clear to me why the items on nervousness, difficulty relaxing, restlessness, irritability and anxiety are emphasised with regard to their somatic component, in contrast to the corresponding items of the PHQ-9. This subdivision of the GAD-7 seems somewhat arbitrary to me. The authors are asked to reconsider their argumentation and to check whether this separation is really necessary.

As the authors correctly note in the discussion, a criterion-standard instrument for generalized anxiety is missing. The criterion standard in this study is the GAD-7 score itself, so that the items are correlated with themselves as part of the GAD-7 score when determining diagnostic accuracy. An external and thus independent criterion standard is missing. The authors might argue why they believe that evidence of the cut-off value of the GAD-7 can be derived from this partial self-correlation, for which external criterion standards are usually used. I think, the rationale for this should be presented in a differentiated manner in the methods section and critically discussed in the discussion section.

It is not clear to me why the diagnostic accuracy of the individual items of the GAD-7 is calculated. After all, the aim of clinical application is to ensure the accuracy of the entire GAD-7 scale or the GAD-2 short form, not the accuracy of individual items. The fact that a scale consists of items of varying difficulty is actually desirable in scales! In my opinion, if the validity of parts of the GAD-7 is to be examined, then it should be the validated short form of the GAD-7, not in the individual items. At the very least, the authors might provide a better explanation of the rationale for this approach.

MINOR:

In the methods section, a brief explanation should be given as to why the data are not very recent (2018); in the discussion, it could be discussed whether this could have an impact on the results.

Results section (3.1): The GAD-7 is not used to measure prevalences! It only measures the frequency of screening diagnoses!

There are high frequencies for anxiety disorders at the cut-off scores of 7 and 8 (22 and 28%). Are these high prevalences clinically plausible? In my opinion, this should be discussed.

Tab. 3 and Tab. 4: Here I would recommend writing out the items – this makes it easier for the reader who does not know the individual items by heart to find his way around.

Conclusions: The conclusion that different response patterns were found for the somatic-emotional and cognitive items is not valid, as these differences have not been statistically tested.

6. PLOS authors have the option to publish the peer review history of their article (what does this mean?). If published, this will include your full peer review and any attached files.

Reviewer #1: No

Reviewer #2: No

---

## [Author Response · Author response to Decision Letter 0]

21 Nov 2024

Response to Reviewers

Heidelberg, 21.11.2024

Dear Reviewers,

Thank you very much for your thorough and insightful comments, which have significantly helped us improve our work. Based on your comments, we made substantial revisions, particularly to the Discussion, but also in the Methods section and the Introduction, to enhance clarity and rigor. Please find below our responses to the individual points raised by you in your comments.

During the revision process, we also identified and corrected an error in Table 2 (“Demographic data, clinical data and prevalence of GAD depending on different GAD-7 cut-offs”): the stand-ard deviation previously reported as 9.21 has been corrected to 6.21. 

We believe these revisions have significantly strengthened the manuscript and hope it is now suit-able for acceptance by PLOS ONE. 

Thank you again for your time and consideration. We look forward to your feedback. 

Kind regards, 

Miriam Grapp (on behalf of all authors)

Answers to Academic Editor / Journal requirements:

Answer:

We have checked that our manuscript complies with the PLOS ONE style guide and revised the formatting of the level 1 and level 2 headings, the citation of figures in the text, the file names for Fig1 accordingly.

2. We note that you have indicated that there are restrictions to data sharing for this study. For studies involving human research participant data or other sensitive data, we encourage authors to share de-identified or anonymized data. However, when data cannot be publicly shared for ethical reasons, we allow authors to make their data sets available upon request. 

Answer:

Thank you for this important remark. After consultation with the ethics committee of the Univer-sity of Heidelberg, we have uploaded a minimal anonymized data set necessary to replicate your study findings on figshare.com. 

We have updated our Data Availability statement in the submission form as follows:

“The dataset generated for this study can be found in the Figshare data repository: https://figshare.com/articles/dataset/Screening_for_anxiety_in_patients_with_cancer_diagnostic_accuracy_of_GAD-7_items_considering_lowered_GAD-7_cut-offs/27880509/1?file=50690409 (https://doi.org/10.6084/m9.figshare.27880509.v1)” 

Answers to Reviewer #1

1. Expand on Methodological Limitations: While the study mentions the lack of validation with another diagnostic measure for anxiety disorders, this limitation could be explored further. Clarifying how the absence of a clinical interview or additional questionnaires might affect the accuracy of the GAD-7 would strengthen the discussion of limitations and transparency.

Answer:

Thank you for mentioning this important issue. We agree with the reviewer's comment and are aware that the lack of an objective criterion for validating GAD-7 scores is a major limitation of our study. We have taken this comment very seriously and have expanded on this aspect in the discussion section (see “Limitations”) as follows:

“Structured or semi-structured interviews are considered the gold standard for diagnosing anxiety disorders. However, this was not possible in our study, which focused on a large data set. Data were sampled in the context of routine care, where interviews would be too time consuming. It is important to understand that the GAD-7 is simply a screening tool to identify generalized anxiety disorder, but cannot replace a clinical diagnosis. […]

However, the absence of an additional criterion for evaluating the GAD-7, e.g., HADS as a sec-ond screening tool or an interview, results in the criterion standard for this study being the GAD-7 score itself. Thus, the predictor and criterion are not completely independent of each other, which potentially bears the risk that sensitivity and specificity will be overestimated or biased. Therefore, sensitivity and specificity analyses should be interpreted with caution. Further research should therefore incorporate clinical interviews with a sufficiently representative subsample of cancer patients ensure consistency with the GAD-7 results.”

2. Consider Including TNM Classification: The study could greatly benefit from incorporating the TNM classification of cancer severity. This would allow for a more detailed analysis of how anxiety levels might correlate with the stage of cancer, offering deeper insights into the physical and psy-chological interaction.

We completely agree with the reviewer that the study would undoubtedly benefit from incorpo-rating the TNM classification. However, it should be noted that the results are based on a routine questionnaire used in clinical care. This questionnaire is designed to screen for psychological dis-tress, and it also collects some socio-demographic variables. Unfortunately, the TNM classifica-tion is not included and is therefore not available to us. We have already mentioned this in the Discussion section (Limitations) and have now expanded the corresponding passage to include the reviewer's suggestions:

“As our data are based on the analysis of a routine questionnaire used in clinical care that focuses on the assessment of psychological distress, no TNM classification was available in the present dataset. Therefore, it was not possible to directly correlate GAD-7 scores with physical disease severity or to perform subgroup analyses. Information on TNM status would allow a more de-tailed analysis of the possible correlation between anxiety and the cancer stage, providing a more comprehensive understanding of the relationship between physical and psychological symptom dimensions.”

Furthermore, we have added the following passage in the conclusion section to highlight the im-portance of this aspect for future research:

“It would be beneficial for future research to consider the stage of the disease, as indicated by the TNM classification. Incorporating the TNM classification into analyses could enhance future re-search by clarifying the relationship between anxiety symptoms and cancer stage, and by deepen-ing insights into the interplay of physical and psychological symptoms.”

3. Elaborate on the Clinical Implications: The conclusion suggests that a lowered GAD-7 cut-off is beneficial for oncological settings. It would be helpful to include more practical guidelines for cli-nicians on how this change might influence decision-making in clinical practice, particularly re-garding referrals to psychosocial support services.

Thank you for highlighting this important aspect. We have now added the following section to the Discussion section (under “Clinical implications”):

“Lowering the GAD-7 cut-off would lead to earlier identification of anxiety symptoms in cancer patients, which is crucial given their vulnerability to psychological distress. The main aim of a lowered GAD-7 cut-off is to miss as few patients as possible with a potential anxiety disorder and to give as many as possible access to psycho-oncological interventions. This would imply an increase in the number of patients flagged for potential anxiety disorders. These patients should be referred to the psycho-oncological service at an early stage in order to verify the diagnosis of an anxiety disorder in a two-stage process (preferably with a clinical interview).”

4. Clarify the Role of Somatic-Emotional and Cognitive Items: The differentiation between somat-ic-emotional and cognitive items is one of the unique strengths of the study. However, the rationale behind this division could be more fully explained…particularly how this distinction is relevant to the cancer population compared to other populations with anxiety disorders.

We very much appreciate your positive feedback, in which you mention the distinction between somatic-emotional and cognitive aspects as one of the unique strengths of the study. Furthermore, we acknowledge that the rationale for this division is not elaborated in sufficient detail in our manuscript. Therefore, we have added the following paragraph to the Introduction to make the reasons for the differentiation between somatic-emotional and cognitive items more comprehensi-ble to the reader:

„The GAD-7 also includes items that might be assigned to a more physical (i.e. somatic) symptom dimension (e.g. ‘trouble relaxing’) and items that might be assigned mainly to a cognitive symp-tom dimension (e.g. ‘not being able to stop/control worrying’). However, there is no such clear and explicit distinction for GAD-7 items as for example in depression screening with PHQ-9 [24] and a discussion about the relevance of somatic items in the identification of anxiety disorders in cancer patients has not yet emerged. Therefore, one aim of the present study is to find out wheth-er a distinction between a somatic-emotional and a cognitive symptom dimension could be rele-vant in the screening of generalized anxiety disorder in patients with cancer.”

Furthermore, we have expanded on the Methods section and described in more detail our classifi-cation process.

“To investigate whether a distinction between a somatic-emotional and a cognitive symptom di-mension might be relevant in screening for generalized anxiety disorder in cancer patients, we needed to classify the GAD-7 items into these two dimensions. To the best of our knowledge, such a categorization did not exist before and therefore there was no preliminary work that we could refer to. We therefore used expert consensus for the categorization: seven raters (clini-cal/scientific experts; psycho-oncologists, physicians) independently assigned each item to one the two symptom dimensions. The classification was determined by the level of agreement between the raters, with items assigned by majority vote when full consensus was not achieved. The result-ing categorization used for our analysis is shown in Table 1.”

5. Strengthen Data Availability Statement: Although the study provides a link to a Figshare reposi-tory for data sharing, it's important to ensure that the repository is fully accessible and includes detailed instructions for others who may want to replicate or further explore the analysis. A clear description of what data is available and how it can be accessed is crucial for PLOS ONE's em-phasis on open science.

Thank you for this important information. The editor also pointed out the weaknesses in our Data Availability Statement. After consultation with the Ethics Committee of the University of Hei-delberg, we have uploaded an anonymized dataset to figshare.com. Thus, the data set we used for our study is now fully accessible. Furthermore, we have revised the labeling of each variable in the dataset so that the use of the variables is clearly understandable for others who may want to replicate or further explore the analysis.

We have updated our Data Availability statement in the submission form as follows:

“The dataset generated for this study can be found in the Figshare data repository: https://figshare.com/articles/dataset/Screening_for_anxiety_in_patients_with_cancer_diagnostic_accuracy_of_GAD-7_items_considering_lowered_GAD-7_cut-offs/27880509/1?file=50690409 (https://doi.org/10.6084/m9.figshare.27880509.v1)” 

6. Consider Exploring Gender Differences further: The study briefly mentions that the sample is predominantly female. Given that anxiety presentations might differ by gender, a deeper analysis of how GAD-7 scores differ between male and female cancer patients could add an important dimen-sion to the findings.

Thank you for this valuable comment. Based on the reviewer's comments, we realized that this paragraph needs to be elaborated on in the Discussion section, both to explain why we did not focus on it in this study and to ensure that the relevance of the issue is not overlooked. Moreover, we consider the reviewer's recommendation a great suggestion for future research and have there-fore included the following this aspect in the Conclusion section. 

Based on this reviewer's comment, we made the following additions to the manuscript:

(1) Discussion/Limitations:

“Moreover, we did not analyze and interpret our results on a sex or gender-specific basis although there is considerable evidence that there are gender differences in the presentation of anxiety symptoms in cancer patients. In particular, recent studies show that female patients are more anx-ious than male patients across all tumor entities [38]. Given the complexity of our research ques-tion, we chose to limit our scope. We did not address gender differences in anxiety disorders and in the presentation of anxiety symptoms. This could slightly restrict the generalizability of our findings.”

(2) Conclusion: 

“Another important aspect for further research is a deeper analysis of how GAD-7 scores and the extent of different anxiety symptoms differ between male and female patients with cancer. Given the known gender differences regarding the suffering of anxiety symptoms in oncological pa-tients, this could add an important dimension to our findings.”

Answers to Reviewer #2

The manuscript by Grapp et al. is an original work that examines the diagnostic accuracy of the items of the GAD-7 scale and, based on this, draws conclusions about the ideal cut-off value of the GAD-7 in cancer patients. It is therefore a clinically relevant and interesting work. The manu-script, which is well written in terms of style, comes from a renowned group in the field of psycho-oncology. Although the manuscript is important, it raises some questions that the authors might consider in the context of a revision:

Answer:

Thank you very much for your encouraging feedback on our manuscript. We are grateful for your recognition of the clinical relevance and importance of our work, as well as your kind words re-garding its style and our contributions to the field of psycho-oncology.

1. Although GAD 7 is named after generalised anxiety disorder, it is also used to screen for other anxiety disorders. In the manuscript, the authors only refer to the accuracy with regard to general-ised anxiety disorder. At the very least, the method section should include a justification for not also examining the other anxiety disorders. If necessary, this should be mentioned as a limitation in the discussion.

Answer:

Thank you for this important comment. We have now provided a more detailed explanation in the Methods section and believe it may not be necessary to further discuss this issue in the Limita-tions section of the Discussion. 

We added the following paragraph to the Methods section:

“Although initially developed as a screening tool for Generalized Anxiety Disorder, the GAD-7 also demonstrates acceptable sensitivity and specificity for identifying other anxiety disorders (Panic Disorder, Social Anxiety, and PTSD) [15]. In the present study, however, the focus was specifically on the symptoms of generalized anxiety disorder. Consequently, the diagnostic accu-racy of the GAD-7 was examined solely in relation to generalized anxiety disorder, and other anxiety disorders were not considered.”

2. While a clear distinction is made in the diagnosis of depressive disorders between somatic and non-somatic items (e.g. somatic items of the PHQ-9: ‘feeling tired or having little energy’ or ‘poor appetite or overeating’), this distinction appears not really comprehensible for the GAD-7. It is not clear to me why the items on nervousness, difficulty relaxing, restlessness, irritability and anxiety are emphasised with regard to their somatic component, in contrast to the corresponding items of the PHQ-9. This subdivision of the GAD-7 seems somewhat arbitrary to me. The authors are asked to reconsider their argumentation and to check whether this separation is really necessary.

Answer:

We understand your concerns at this point. To the best of our knowledge, this separation has not been addressed in previous studies. As such, the categorization of the items into somatic-emotional and cognitive aspects—based on the approach used for the PHQ-9 items—represents one of the innovative aspects of our study. However, we agree that this categorization can only be considered as a model.

Based on the feedback from both reviewers, we have also reali

---

## [Editor Report · Decision Letter 1]

17 Dec 2024

Screening for anxiety in patients with cancer: diagnostic accuracy of GAD-7 items considering lowered GAD-7 cut-offs

PONE-D-24-37266R1

Dear Dr. Grapp,

We’re pleased to inform you that your manuscript has been judged scientifically suitable for publication and will be formally accepted for publication once it meets all outstanding technical requirements.

Kind regards,

Stephan Doering, M.D.

Academic Editor

PLOS ONE
---

## [Editor Report · Acceptance letter]

3 Jan 2025

PONE-D-24-37266R1 

PLOS ONE

Dear Dr. Grapp, 

I'm pleased to inform you that your manuscript has been deemed suitable for publication in PLOS ONE. Congratulations! Your manuscript is now being handed over to our production team.

Kind regards, 

on behalf of

Professor Stephan Doering 

Academic Editor

PLOS ONE